# Exploring psychiatrists' perspectives on supporting parents with mental health Challenges: A mixed-methods study

Mireille Jasmin[1,2], Geneviève Piché[1,2,3]*, Aude Villatte[1,2,4], Andrea Reupert[5], Marie-Ève Clément[1], Anne Dorothee Müller[6], Marianne Fournier-Marceau[1], Darryl Maybery[7], Marie-Hélène Morin[8], Stéphane Richard-Devantoy[9,10]

1 Université du Québec en Outaouais, Département de psychoéducation et de psychologie, Saint-Jérôme, Quebec, Canada, 2 Centre de recherche universitaire sur les jeunes et les familles, Quebec, Quebec, Canada, 3 Réseau de recherche en santé des populations du Québec, Montreal, Quebec, Canada, 4 Laboratoire Psychologie de la Socialisation: Développement et Travail, Université Toulouse Jean Jaurès, Toulouse, France, 5 School of Educational Psychology and Counselling, Faculty of Education, Monash University, Clayton, Victoria, Australia, 6 Child and Adolescent Mental Health Center, Copenhagen University Hospital – Bispebjerg and Frederiksberg, Copenhagen, Denmark, 7 Department of Rural and Indigenous Health, Monash University, Clayton, Australia, 8 Département de travail social, Université du Québec à Rimouski, Rimouski, Quebec, Canada, 9 McGill University, Department of Psychiatry and Douglas Mental Health University Institute, McGill Group for Suicide Studies, Montréal, Quebec, Canada, 10 Saint-Jérôme Hospital, Saint-Jérôme, Quebec, Canada

* genevieve.piche@uqo.ca

## Abstract

### Background

Parenting responsibilities can be particularly challenging for patients receiving mental health services, often resulting in a range of negative impacts on children. Incorporating a family-focused approach into the usual care of parents with mental illness has been recommended to promote patient recovery while supporting the well-being of children and the entire family unit. This study aimed to document the family-focused practices undertaken by psychiatrists working with parents who have a mental illness and to explore potential facilitators and barriers to these practices.

### Methods

A sequential explanatory mixed-method design was used, combining an online survey and individual interviews. Family-focused practices were reported by 27 psychiatrists through the French version of the Family-Focused Mental Health Practice Questionnaire. Follow-up qualitative individual interviews were conducted with 5 psychiatrists. Item-by-item analysis of the quantitative data was performed, followed by a thematic analysis of the qualitative data, integrating findings from both sources.

**Data availability statement:** Data cannot be shared publicly because participants did not provide consent for open data sharing, the dataset contains potentially identifying and sensitive information, and the Research Ethics Committee has restricted public access to the dataset. Data are available from the Université du Québec en Outaouais Ethics Committee (contact via comite.ethique@uqo.ca) for qualified researchers who meet the criteria for access to confidential data.

**Funding:** GP received funding from the Social Sciences and Humanities Research Council of Canada (SSHRC) (#435–2020–1092). GP also received financial support from the Réseau de Recherche en Santé des Populations du Québec (RRSPQ). The funders of the study had no role in the study design, data collection, data analysis, data interpretation, or writing of the manuscript.

**Competing interests:** The authors declare that they have no competing interests.

## Results

Although psychiatrists acknowledge their patients' parenting role, most are reluctant to provide further support. Key barriers to family-focused practice include the predominantly individual-focused nature of psychiatric care, stigma, consent issues, and limited collaboration between adult and child services. Facilitators include psychiatrists' professional autonomy, personal experience, and confidence in conducting family meetings.

## Conclusion

Psychiatrists can play a pivotal role in identifying, acknowledging, and providing appropriate support to parents with mental illness and their families, including children. Developing comprehensive guidelines and targeted training is essential to equip psychiatrists with effective strategies for addressing parenting challenges in patients with complex mental health issues. Additionally, psychoeducational resources for children should be incorporated. Implementing these initiatives may lead to more compassionate, targeted care and improved outcomes for parents and their families.

## Background

Many psychiatric patients are parents, with estimates ranging from 20–38.5% in various services [1,2]. Thus, an important aspect of psychiatric care and recovery involves patients who are parents. The mental health of these parents affects not only themselves but also their children's development and family well-being, often impairing parents' ability to meet children's emotional and educational needs [3]. Parents may experience guilt and concern about these impacts [4]. Identifying, acknowledging, and supporting patients' parenting role is key to recovery, as embedding parenting into treatment encourages adherence to medication and motivates parents to "get better" for the sake of their children [5], while also boosting their confidence in parenting [6]. Parenthood also fosters identity, connectedness, and hope, crucial elements of recovery [7–9].

Children of parents with mental illness face increased risks of mental health issues [10–12] and psychosocial challenges [13]. They often take on caregiving responsibilities for their parents [14] while struggling with the emotional "rollercoaster" of their family life [15]. These children express a strong need to receive information from healthcare professionals about their parents' conditions [16] and to discuss their experiences [17].

Thus, there is a need for a family-focused approach in mental health services that goes beyond treating adult patients alone by addressing the needs of the entire family. Family-focused practices emphasize a whole-family perspective, fostering collaborative partnerships, being family-centered, and supporting families in the care and development of individuals [18]. This approach promotes a systemic view,

recognizing the reciprocal effects of symptoms on parenting and family functioning [19,20]. Such practices and family-oriented interventions have been shown to improve outcomes for individuals with mental illness, including reduced relapse and hospitalizations rates [21–23]. They also enhance parental efficacy [24], strengthen parent–child relationships, promote resilience and coping in children [25–27] and reduce the risk of intergenerational transmission of mental illness [28].

Canadian policy frameworks increasingly emphasize family involvement in mental healthcare. The Mental Health Strategy for Canada (2019) encourages institutions to engage families in prevention, diagnosis, treatment, service planning, and evaluation. This represents a local and global shift toward a relational recovery model and a "family-centered" approach, in contrast to a traditional individualized, symptom-centered model of care [29]. However, the current mental healthcare system in Canada faces significant challenges, including long wait times, resource shortages, and disparities in access, particularly in rural and underserved areas. The separation between specialized mental health services and general primary care, as well as between child and adult services [30], has created barriers to providing timely, integrated care, which is considered a facilitator for adopting a more family-focused practice [31].

In collaboration with other professionals, psychiatrists, who play a central role in managing complex mental illnesses [32,33], are encouraged to involve families not only in treatment planning to support patient recovery but also in providing them with resources and support to help them cope with caregiving challenges [34]. Given the key role they play within mental health services [32,33], it is important to ascertain how psychiatrists support families in navigating mental health challenges. In two large-scale quantitative studies, psychiatrists reported providing higher levels of support to patients' families and children than other professional groups [35,36], including providing information on mental illness to children, parenting advice to parents, and referral resources. However, findings from two qualitative studies reveal a more limited involvement. In Laser et al study (38), psychiatrists mainly addressed parenting concerns in two contexts: arranging child care arrangements during hospital admissions and discussing potential pregnancies in relation to psychiatric medication [37]. In Cognard & Wendland, psychiatrists acknowledged the importance of the mother–child relationship, but focused primarily on assessing maternal mental health and caregiving capacity, rather than offering direct family support [38]. Across both studies, concerns about confidentiality, lack of training, and unclear protocols were cited as key barriers to engaging in family-focused practices.

In Canada, as elsewhere, addressing the needs of family members is considered essential [39]. Psychiatrists are well-positioned to offer this support, making their involvement a critical opportunity to better integrate family-focused care. However, few studies have explored how psychiatrists perceive their role in addressing the parenting needs of patients with mental illness and the strategies they might employ to support these families. Prior research highlights a discrepancy between psychiatrists' self-reported levels of family support, as reported in quantitative studies, and the relatively limited, assessment-focused involvement observed in qualitative research. This discrepancy highlights the disconnect between what psychiatrists report in structured surveys and how they describe their involvement with families in qualitative interviews, which may also imply that quantitative and qualitative methods assess different facets of family-focused practices. Few studies have specifically examined psychiatrists' practices. Most studies focus on other mental health professionals, such as nurses, pediatricians and family physicians [40–42]. This study aims to address these gaps by investigating first how psychiatrists in Quebec, Canada, support parents with a mental illness and their families, and second, the factors that encourage or hinder their engagement in family-focused practices. The findings will deepen our understanding of psychiatrists' contributions to supporting families in navigating mental health challenges, with recommendations provided to strengthen family-focused approaches to psychiatric care.

## Methods

This study used a sequential explanatory mixed-method design. This approach was used to develop a nuanced description and interpretation of the phenomenon, in which qualitative data further contextualize and enrich the quantitative results [43]. Phase I involved the completion of the *Family-Focused Mental Health Practice Questionnaire*

(FFMHPQ-FR) [44], which aims to document psychiatrists' family-focused practices. Phase II included individual interviews with selected Phase I participants to explore psychiatrists' perspectives on their family-focused practices and the factors that support and hinder them. Phase III of this study sought to integrate the quantitative and qualitative data collected in Phases I and II. Guidelines for the Good Reporting of A Mixed Methods Study (GRAMMS) [45] were used.

This study was approved by the Research Ethics Board of the Université du Québec en Outaouais (#2021–1167) and the Comité d'Éthique de la Recherche Sectorielle en Santé des Populations et Première Ligne du CIUSSS de la Capitale-Nationale (# MP-13-2021-2135). All participants signed an online information and consent form before completing the online questionnaire and before participating in the interviews.

## Recruitment

Recruitment for Phase I was conducted across all regions of the Province of Quebec, Canada, from 15/03/2021–08/02/2022. Various strategies were used (e.g., emails to managers of main workplaces and professional associations, advertisements on social media). Participants were required to meet the following criteria: 1) be psychiatrists; 2) currently work with adults with mental illness; 2) have direct contact with clients; and 3) be fluent in French. The only exclusion criterion was working exclusively with children with mental illness. Recruitment for Phase II took place from 15/06/2023–15/08/2023. A few months after Phase I, the psychiatrists who had participated in that phase were invited by email to take part in Phase II. Of the 27 psychiatrists contacted, five agreed to participate (two women and three men).

## Participants

Twenty-seven psychiatrists completed the online survey (Table 1). Gender distribution was equal, and most participants (88.5%) were born in Quebec. The average age was 47.1 years (range 29–81), with an average of 14.2 years of experience (range 1–48). All psychiatrists worked in hospital settings (100.0%), providing second- or third-line services such as psychiatric emergencies, inpatient care, or outpatient clinics. On average, they estimated that 35.7% of their clients had parenting responsibilities (range: 5–75%), aligning with rates reported in similar contexts [46]. In Phase II, five participants took part in individual interviews (two women and three men), with ages ranging from 37 to 81 years and an average of 22 years of experience.

## Measures

The French version of the FFMHPQ [47] (FFMHPQ-FR) were used to measure psychiatrists' practices, knowledge and confidence when working with families that have a parent with a mental illness. The FFMHPQ-FR, previously translated and validated, showed good psychometric properties [44]. The three subscales used in this study were 1) *Family-focused practices*; 2) *professionals' perceived skills, knowledge, and confidence*; and 3) *professionals' attitudes and beliefs toward FFP*. The participants rated their answers on a 7-point Likert scale (from 1 = *strongly disagree* to 7 = *strongly agree*). The questionnaire took 10–15 minutes to complete and included questions about participants' professional background and work context (e.g., experience, workplace, and proportion of clients with children).

Semi-structured online interviews were conducted via Microsoft Teams using a guide developed from the research aims, Phase I preliminary results, and the literature. The interview guide (see Supplementary File) explored psychiatrists' experience working with parents with mental illness (e.g., *What is it like for you to work with adults who have a mental illness and also have a parenting role? In your opinion, what is your role in accompanying parents with mental illnesses? How, if at all, is it different from working with adults who do not have children?*). Interviews lasted 38–66 minutes (average, 47 minutes) and were audio-recorded.

 

**Table 1. Participants' sociodemographic and professional background characteristics.**

| | Psychiatrists (*n*=27) |
|---|---|
| **Continuous variables** | Mean (SD) |
| Age, *years* | 47.72 (16.17) |
| Experience in current position, *years* | 14.15 (11.99) |
| Reported proportion of clients having children, % | 35.73 (20.10) |
| **Categorical variables** | N (%) |
| Gender | |
| Women | 13 (48.1) |
| Men | 13 (48.1) |
| Place of birth | |
| Quebec | 23 (88.5) |
| Elsewhere | 3 (11.5) |
| Principal workplace | |
| Medical clinics of medical family groups (GMF) | 0 (0.0) |
| Hospital or university hospital centers | 27 (100.0) |
| Local community service centers (CLSC) | 0 (0.0) |
| Private practice | 0 (0.0) |
| Basis of employment | |
| Full-time | 25 (92.6) |
| Part-time | 2 (7.4) |

Note: Sample size varied due to missing values.

## Data analyses

Participants could choose "refuse to answer" or "not applicable", which were coded as missing answers and treated with mean value imputation by item [48]. Across all 38 items, the rate of valid responses was 99.0%. Quantitative analysis was performed using SPSS 27 software (IBM Corp., 2020). Basic assumptions for the parametric techniques were verified (e.g., normal distribution) [49]. One sample t-tests compared item scores to subscale means, identifying items significantly above or below average. Higher scores indicated more family-focused practices or facilitating factors, whereas lower scores reflected fewer family-focused practices or hindering factors. Statistical significance was set at $P < 0.05$.

Qualitative data were analyzed using NVivo 14 through thematic content analysis, based on Braun and Clarke's six phases [50]: familiarization, coding, theme generation, review, refinement, and reporting. Weekly team meetings ensured consensus on codes and themes.

Following the qualitative data analysis, three co-authors conducted an integration phase to combine the quantitative and qualitative findings. This process was presented via a joint display figure, which highlights new insights that go beyond those revealed by the quantitative and qualitative results individually.

## Results

### Quantitative findings

Table 2 presents the means and confidence intervals of the scores of the 27 psychiatrists regarding their level of family-focused practices, skills and knowledge, and attitudes toward family-focused practices. It also presents items that are significantly above and below the mean.

**Table 2. Item-by-item analysis.**

| | | Participants (n = 27) | |
|---|---|---|---|
| Items by subscale | | M [95% CI] | Δ² |
| ***Family-focused practices*** | | **4.81 [4.51-5.12]** | |
| 1 | In my practice, I instinctively check if the adult with a mental illness I'm working with has a child or children (aged 0–25 years). | 6.81 [6.45-7.17] | + 2.00*** |
| 2 | I regularly check whether parent-users are able to meet the socio-emotional needs of their children. | 6.11 [5.83-6.39] | + 1.30*** |
| 3 | I provide written materials (e.g., education. information) about parenting to parent-users. | 2.67 [2.00-3.34] | − 2.15*** |
| 4 | I regularly provide information (including written materials) about mental health issues to the children of parent-users with whom I work. | 4.19 [3.58-4.80] | − 0.63 |
| 5 | I do meet with parents and families in the course of my employment (not family therapy). | 6.00 [5.64-6.36] | + 1.19*** |
| 6 | I regularly discuss strategies with parent-users to strengthen their parenting practices or better meet the needs of their children. | 4.59 [3.98-5.20] | − 0.22 |
| 7 | I regularly refer the children of parent-users to specific support services. | 4.59 [3.97-5.21] | − 0.22 |
| 8 | I regularly refer parent-users to parent-related programs (e.g., parenting skills). | 3.78 [2.97-4.59] | − 1.04* |
| 9 | I regularly refer the partners of parent-users to specific support services. | 4.58 [3.97-5.19] | − 0.23 |
| ***Skills, knowledge and confidence in FPP*** | | **4.94 [4.62-5.27]** | |
| 11R | I am not confident working with parent-users about their parenting skills. | 4.76 [4.08-5.44] | − 0.18 |
| 13 | I am able to determine the developmental progress of the children of parent-users with whom I work. | 3.07 [2.35-3.79] | − 1.87*** |
| 14 | I am knowledgeable about how parental mental illness impacts on families. | 6.19 [5.81-6.57] | + 1.24*** |
| 15 | I am able to assess the importance that parent place on their children participation in social and recreational activities. | 5.00 [4.46-5.54] | + 0.06 |
| 17R | I do not have enough confidence in my professional skills to work with the families of parent-users. | 5.03 [4.42-5.64] | + 0.08 |
| 20 | I am able to assess the extent to which the manifestations of mental illness of parent-users have an impact on their children. | 5.41 [4.89-5.93] | + 0.46 |
| 21R | I do not have the skills to help parent-users recognize the possible impact of their mental illness on their children and families. | 5.59 [5.05-6.13] | + 0.65* |
| 22 | I am able to determine the importance that parent-users place on maintaining good relationships between their children and other family members. | 5.59 [5.18-6.00] | + 0.65** |
| 28R | I have no experience in working with the children of parent-users. | 4.89 [4.22-5.56] | − 0.06 |
| 29R | I am not able to assess the importance that parent-users place on the quality of the support network (outside the family) of their children. | 4.63 [4.00-5.26] | − 0.31 |
| 33 | I am skilled to support parent-users in promoting the well-being of their children. | 4.96 [4.43-5.49] | + 0.02 |
| 36R | I do not have enough confidence in my professional skills to work with the children of parent-users. | 5.00 [4.36-5.64] | + 0.06 |
| 37 | I know the main techniques that parent-users could use to promote the well-being of their children. | 4.15 [3.51-4.79] | − 0.80* |
| ***Attitudes and beliefs toward FFP*** | | **5.80 [5.39-6.22]** | |
| 24 | No matter the level of severity or type of illness of the parent-users. use of a family centered practice is always relevant. | 5.85 [5.34-6.36] | + 0.05 |
| 25R | Use of a family-centered practice is irrelevant when there is no risk of violent or suicidal behaviour in a parent-user. | 5.93 [5.22-6.64] | + 0.12 |
| 27 | Adopting a family-centered practice is always relevant, regardless of the age of the children of the parent-users. | 5.63 [5.00-6.26] | − 0.17 |

*Note.* Scores range from 1 = *Strongly disagree* to 7 = *Strongly agree*. R indicate items that were recoded and must be interpreted as such: a high score. Δ indicate the difference between the item mean and the overall mean score for the group of participants. *p < 0.05; **p < 0.01; ***p < 0.001.

The results from the family-focused practices subscale showed large variability across items. Psychiatrists reported that they 'agree' to 'strongly agree' about checking if their patient has a child and checking if their patient can meet their children's needs but reported that they 'disagree' to 'slightly disagree' in providing written materials about parenting to their patients.

The results regarding psychiatrists' self-reported skills, knowledge, and confidence in family-focused practices also revealed variability. While psychiatrists generally agree that they are skilled and knowledgeable about the impact of parental mental illness on families and the importance of social support networks, they slightly disagreed about their knowledge of key techniques that patients could use to promote their children's well-being and their own ability to assess the overall development of their children.

The items on the *'Professionals' attitudes and beliefs toward the FFP* subscale did not vary significantly. Psychiatrists generally agreed on all the items pertaining to the relevance of family-focused practices.

## Qualitative findings

The participants confirmed that screening for parental status was integrated into their usual care and was systematically performed, highlighting its significance as a source of information. Psychiatrists also described assessing parents' capacity to respond to their children's needs. Nonetheless, although they mentioned their willingness to address parenting concerns, this focus was underlined as limited, especially when parents were not actively engaging in or furthering discussions on their parenting role.

Psychiatrists also reported making concerted efforts to include family members in their sessions, particularly during evaluation meetings and discharge planning, and reported that part of their role was to listen to their patients' relatives. At the same time, psychiatrists specified that family meetings rarely include children (Table 3).

### Barriers to family-focused practice

Some professionals expressed hesitation about involving children, citing concerns about the appropriateness of discussing parents' psychological difficulties in their presence. This may reflect underlying beliefs about the unsuitability of family-focused approaches in such contexts, with discomfort rooted in confidentiality issues and the sensitive nature of psychiatric information.

Psychiatrists noted that other professional groups might be better suited for family-focused interventions, emphasizing that although they valued working with the family, "…it is not my professional responsibility to do that work' (Psychiatrist 4). Most participants acknowledged their predominant focus on individual-centered care, which often took priority over broader family-oriented approaches.

While psychiatrists generally felt confident in their skills, many reported feeling ill-equipped and lacking specific training in discussing parenting techniques and child development, which they belief would be beneficial to improve their practice.

Participants also identified a significant structural barrier in the form of limited collaboration between adult and child services. This fragmentation hindered their ability to effectively refer patients' children to appropriate support and constrained their understanding of the family's broader needs.

Some barriers related to patients and their families were underlined. Several participants noted that they rarely met their patients' children, often due to the children's unavailability because of school or because parents "have often lost custody of their children." (Psychiatrist 4). Additionally, parental self-stigma emerged as a significant obstacle, with some psychiatrists reporting that patients were hesitant to disclose information about their children out of "fear of consequences, fear of legal consequences, or simply fear of being judged". (Psychiatrist 4).

### Facilitators of family-focused practice

Participants emphasized the importance of recognizing the parental role of their patients and involving families in patient care, highlighting the significant benefits of a family-focused approach. They reported feeling confident in their ability to work with patients' families, attributing this in part to the training they had received.

Despite challenges related to patient reluctance to involve family members, psychiatrists identified several professional skills, such as negotiation, that helped them navigate and overcome parental hesitancy. Additionally, some participants noted that their own personal experience as parents enhanced their capacity to connect with and support other parents.

**Table 3.** Integrative results from Phases I and II.

| Quantitative results | Qualitative results | Integrated results |
|---|---|---|
| **Family-focused practices** | | |
| **Screening and evaluation** | | |
| Item 1 = In my practice, I instinctively check if the adult with a mental illness I'm working with has a child or children. *High score* | • **Practice that is integrated into usual care and systematically done at the beginning of the evaluation** *"If we want to care for someone and have an idea of who they are, knowing if they have children is primordial. It's part of the basic information needed."* (Psychiatrist 4). | Converging results, screening for a patient's family is described as essential work. |
| Item 2 = I regularly check whether parent-users are able to meet the socio-emotional needs of their children. *High score* | • **Ask about how things are going with the children** *"Often when I talk with my patients, I will ask them at some point 'So how's it going with your partner? How's it going with your children?' But you know, I'll admit that my screening isn't any deeper than that. So, if it isn't brought up when I ask, 'How's it going with your children?', then […] However, if they do open up and we talk about it together, then, I'll be able to help them."* (Psychiatrist 2). *"We are interested in the children and we try to touch on the topic, at least the question, is there a need there? Is there a problem? How is it going with the children?"* (Psychiatrist 4). | Converging results, psychiatrists ask about how things are going with the children |
| **Engaging with the family and giving resources** | | |
| Item 5 = I do meet with parents and families in the course of my employment (not family therapy). *High score* | • **Always invite family to the meetings** *"I've always invited them; every time I have a parent, an adult patient or younger, if you want to come with your partner, your children, your family, your parents, you're always welcome, or with a close one or a friend. […] So yes, working with the family."* (Psychiatrist 1). • **Listen to the family member's distress** *"If we want to support the families, we have to listen to them instead of informing them. I start by asking: 'What are you going through? What's going on with you? What is upsetting you? Is there a situation that you never seem to be able to manage?'"* (Psychiatrist 3). • **Family meetings rarely include children** *"It's mostly with the partners that I work with or with the parent of the parent or with another close one who is significant to them."* (Psychiatrist 2). | Converging results, engaging with a patient's family is described as essential work. Yet, invitations to participate does not equal participation, family meetings rarely include children and relatives. |
| Item 3 = I provide written materials (e.g., education, information) about parenting to parent-users. *Low score* <br><br> Item 4 = I regularly provide information (including written materials) about mental health issues to the children of parent-users with whom I work. *Average score* <br><br> Item 7 = I regularly refer the children of parent-users to specific support services. *Average score* <br><br> Item 8 = I regularly refer parent-users to parent-related programs (e.g., parenting skills). *Low score* <br><br> Item 9 = I regularly refer the partners of parent-users to specific support services. *Average score* | • **Listening instead of giving information** *"If we want to support the families, we have to listen to them instead of informing them"*. (Psychiatrist 3). | Converging results, providing material and referral to support seems less important than listening. |

*(Continued)*

**Table 3.** (Continued)

| Quantitative results | Qualitative results | Integrated results |
|---|---|---|
| **Barriers to family-focused practice** | | |
| **Beliefs around FFP** | | |
| *Moderate to high score* for all items on attitudes and beliefs about FFP, no significant difference from the mean | • **Inappropriate**<br>"*The possibly inappropriate aspect of discussing the parents' psychological difficulties in front of his children.*" (Psychiatrist 2). | Diverging results, beliefs of inappropriateness of FFP. |
| **Role perception** | | |
| ø | • **A focus on the patient and individual recovery**<br>"*I try to treat the person as best as possible so that they can use their cognitive and affective capacities and fulfill their parental role. But ultimately, I'm not the one to teach them how to be an adequate parent.*" (Psychiatrist 2).<br>"*…We give medications, we focus on the magical powers of tablets, we are quick quick quick on prescriptions* (Psychiatrist 5).<br>• **Other professionals are seen as more family centered**<br>"*In the biopsychosocial model, we value focusing on family including children, but it is not my professional responsibility to do that work. However, simply paying attention and considering it, yes*". (Psychiatrist 4).<br>"*I'm then only health care provider for many of my patients. Working alone, I do try to assess but I can't pick up everything. So, of course, interdisciplinarity would definitely facilitate my work. (…) If I go with my usual screening, I have so many spheres to assess that my evaluation is more superficial.*" (Psychiatrist 2). | Information only emerged from qualitative phase: psychiatrists report being mainly individual centered.<br><br>Belief that it is the role of other professionals. |
| **Skills, knowledge, and confidence in FFP** | | |
| Item 37 = I know the main techniques that parent-users could use to promote the well-being of their children.<br><br>*Low score*<br>Item 13 = I am able to determine the developmental progress of the children of parent-users with whom I work.<br><br>*Low score*<br>Item 28R = I have no experience in working with the children of parent-users.<br>*Average score* | • **Feeling ill-equipped regarding parenting techniques**<br>"*I have a general idea of the techniques which parents can use to promote their children's well-being but at the same time, has it been taught to me? No. Did I get training on that? No again. Getting trained on the principal techniques could be useful*". (Psychiatrist 2). | A high feeling of confidence and competency is described, except around parenting techniques or child development issues. |
| **Structural barriers** | | |
| ø | • **Lack of collaboration between adult and children services.**<br>"*I don't have any contact with youth-child-family services. They may be working with families I'm also working with, and I might not even be aware of it. Better communication would probably add value*" (Psychiatrist 2). | Information only emerged from qualitative phase. |
| **Patient and family-related barriers** | | |
| ø | • **Unavailability of children and family members**<br>"*My own clientele, my patients are mostly made up of people with psychotic problems that often have children but have often lost custody of their children. So, it's honestly unusual that I meet my patients with their children.*" (Psychiatrist 4).<br>• **Reluctance of the parents to disclose information**.<br>"*Fear of consequences, fear of legal consequences or simply fear of being judged.*" (Psychiatrist 4). | Information only emerged from qualitative phase. |

**Table 3.** (Continued)

| Quantitative results | Qualitative results | Integrated results |
|---|---|---|
| | | |
| **FACILITATORS OF FFP** | | |
| **Attitudes and beliefs** | | |
| **Relevance** | | |
| *Moderate to high score* for all items, no significant difference from the mean | **• Considering families is important**<br>*"From my point of view, when working with the philosophy of recovery, the parental role is an essential part of it. But one must be in the mindset of recovery and not just treating the symptoms."* (Psychiatrist 2)<br>*"I believe that I often save time by meeting the close ones. It really helps to have a better impact."* (Psychiatrist 1). | Converging results on the relevance of family-focused practices. |
| **Skills, knowledge, and confidence in FFP** | | |
| Item 14 = I am knowledgeable about how parental mental illness impacts on families.<br>*High score*<br>Item 21R = I do not have the skills to help parent-users recognize the possible impact of their mental illness on their children and families.<br>*High score* | **• Some training in family intervention enabling confidence**<br>*"Personally, I don't have any difficulties with that because I know how to handle it, and I like doing family meetings."* (Psychiatrist 1).<br>*"The internships [in first psychotic episode's clinics and in personality disorders clinics] motivated us to work in collaboration with the families. Those were two settings where working with families was well integrated. And I think that training is what allows me to include it in my practice as a psychiatrist".* (Psychiatrist 2). | A high feeling of confidence and competency is described. |
| ø | **• Professional skills**<br>*"When I want to meet the relatives, I inform my patient by telling him 'I'll meet your relatives next week. Rest assured, I will not tell them anything you have told me, but it is important that I get to know your family'".* (Psychiatrist 1).<br>*"There is obviously the issue, the obstacle of confidentiality and the parents' refusal. And, honestly, we usually try to work around that refusal, "negotiate", discuss advantages of involving families to try to convince a patient to allow us to talk with one of their family members and bring them to their meetings."* (Psychiatrist 4). | Navigating around confidentiality helps to engage family members |
| ø | **• Personal experience and knowledge**<br>*"That expertise does not come at all from medical school, it comes from the fact that I'm the father of four children and therefore well.... Honestly, when discussing about parenting with my patients, I'm talking about my experience as a father in a certain way.. Of course, there's another part that comes from my training, … but there's a bigger part again, coming from my personal background".* (Psychiatrist 4). | Personal experience with parenting is seen as helpful. |
| **Role perception** | | |
| ø | **Psychiatrists report having high professional autonomy**<br>*"For me, as a doctor, I have a great deal of professional freedom. If I want to involve the family more, nobody's going to stop me."* (Psychiatrist 4).<br>*"I think I may have a bias because I am a doctor, and I am allowed to push a little bit more without reproach."* (Psychiatrist 3). | Information only emerged from qualitative phase; professional autonomy from professional role. |

Professional autonomy also emerged as a key facilitator, providing psychiatrists with the flexibility to tailor their practice. As one psychiatrist remarked, "As a doctor, I have a great deal of professional freedom. If I want to involve the family more, nobody's going to stop me" (Psychiatrist 4).

## Discussion

This study explored how psychiatrists support parents with mental illness and the factors influencing their family-focused care. Results revealed moderate engagement in such practices, contrasting with previous studies reporting higher levels

[35,36]. Nevertheless, our findings reveal considerable variability in psychiatrists' support for patients and families, consistent with prior qualitative research [37].

Primarily, our results suggest that psychiatrists are generally proficient at identifying whether patients have children, an important finding given the commonly observed gaps in recording parental status within mental health services [51,52]. Recognizing parenting status is a fundamental first step in family-focused care and should be routinely integrated into intake assessments across mental health services [53].

However, psychiatrists in our study reported a limited or superficial focus on providing parenting support or directly engaging with patients' children. While they recognized the value of involving family members, including children, this involvement was primarily seen as a means to improve symptom management and recovery, rather than to support the family's understanding or coping with mental illness. This aligns with findings from a recent study of adults who grew up with a parent with mental illness, who felt they were "involved but not included" by professionals, serving as sources of information without sufficient support or inclusion in care decisions [54]. Similarly, psychiatrists here recognized the impact of parental mental illness on families but, consistent with Cognard and Wendland study [38], expressed low confidence in discussing parenting skills or assessing children's developmental needs. Although most reported attempting to address children's needs, such screening rarely extended beyond surface-level engagement.

Furthermore, although some psychiatrists invited family members to meetings, children were seldom included. This echoes results from studies involving various professionals, where concrete support for family members, especially children, was limited [2,52]. A Canadian study from two decades ago similarly found that while service providers acknowledged whole-family needs, few felt equipped to engage all family members effectively [55]. Cognard and Wendland (38) also highlighted a mother-centered approach shaped by limited skills and training, offering minimal direct support for family cohesion or parenting. Such gaps, widely noted among nonphysician professionals as well [56,57,58], underscore an urgent need for resources and training to better equip psychiatrists in addressing child development and parenting concerns during follow-up care for parents with mental illness.

Unlike other studies [36], psychiatrists in our sample did not identify service availability or workload as major barriers. Instead, they emphasized their professional autonomy and independence. Nevertheless, consistent with prior research [37], many expressed reluctance to discuss parental mental illness in front of children due to concerns about appropriateness and confidentiality. Other reasons for limited involvement with patients' children included issues concerning children's unavailability due to school schedules, loss of custody, parental self-stigma, and fear of possible legal consequences.

While some participants took pride in their skills and enjoyed conducting family meetings, others acknowledged their practice remained predominantly individual-centered. This narrow focus represents a significant gap, as addressing parenting can strongly motivate recovery, whereas ignoring parenting stress can harm overall well-being [59]. The emphasis on individual care, which overlooks family dynamics, likely reflects a structural barrier that fragments care between adult and child services rather than addressing the family unit holistically. Growing evidence supports integrated approaches, such as mother-baby units, that yield better outcomes for both parents and children [60,61]. Moreover, although psychiatrists are trained in the biopsychosocial model and recovery care, some feel that "other professionals may be better suited" to support patients' families. As Linden noted [62], psychiatrist training often prioritizes psychopharmacology, with limited emphasis on therapeutic communication, complicating the integration of family support into psychiatric care.

## Implications for practice

Previous research highlights challenges professionals face when discussing parenting and sensible issues with parents who have a mental illness and their families [63,64]. Although psychiatrists in our study routinely identified parenting status, many could benefit from additional training to build confidence and skills in addressing parenting concerns. This learnable skill [65] is essential for helping parents balance mental health and parenting responsibilities, ultimately improving outcomes for both parents and their children. Integrating specific family-focused strategies into clinical training would

better equip psychiatrists to have these sensitive conversations, fostering a more supportive, inclusive environment for families. Evidence shows that training professionals in family-focused care like *Let's Talk about Children* and *The Parenting Well* effectively enhances family-focused practices [59,66].

Moreover, acknowledging psychiatrists' practice limitations underscores the need for a more integrated, multidisciplinary approach to address family issues comprehensively. Psychiatrists should collaborate closely with social workers, psychoeducators, psychologists, nurses and child welfare professionals. Such collaboration allows psychiatrists to concentrate on mental health treatment while ensuring the broader needs of families, especially children, are addressed. Some professionals, such as social workers, may be better positioned to offer direct support to children by responding to their needs, providing information, and engaging parents in their caregiving roles. Nonetheless, psychiatrists' unique role in patient care requires them to remain actively involved in identifying parenting status, assessing parenting concerns, including family members in care plans, and making appropriate referrals to psychosocial or child services. The value of multidisciplinary, interagency, and collaborative care models for families coping with mental illness has been well documented [31,39].

To better support psychiatrists in engaging with families, especially in complex cases involving intersecting family challenges, mental health services could implement interagency family case consultation teams [31]. These teams, comprising experts from various areas (e.g., child services, adult services, community services, youth protection), may provide valuable insights into the complex dynamics families face. Research shows that such interprofessional and interagency support enhances practitioners' confidence in supporting families, improves child safety perceptions, and leads to better clinical outcomes [5,31,67,68].

### Strengths and limitations of the study

A key strength of this study is its focus exclusively on psychiatrists' contributions to supporting families facing mental health challenges, a perspective not previously explored in depth. The mixed-methods design, integrating both qualitative and quantitative data, provides a comprehensive understanding of the topic. However, the study has limitations. The relatively small sample size and overrepresentation of older, more experienced psychiatrists, who typically report higher family-focused practices, may limit the generalizability of findings. Reliance on self-reported data and interviews also introduces potential social desirability bias. Future research could build on these findings by including patient perspectives and reviewing intake records to more objectively verify psychiatrists' family-focused practices.

### Conclusion

While psychiatrists recognize the importance of parenting in mental illness, many hesitate to offer comprehensive support due to barriers like individual-focused practice, stigma, consent challenges, and poor coordination between adult and child services. Facilitators such as professional autonomy, personal experiences, and confidence in conducting family meetings help promote family-focused care. To address these barriers, integrated multidisciplinary approaches and targeted training on parenting challenges and psychoeducation for children are needed. Equipping psychiatrists with these tools can foster family-inclusive care and improve outcomes for parents and their families.

### Acknowledgments

The authors thank all the organizations that helped with the recruitment of participants and all professionals who agreed to take part in this study. We are also grateful for the advice and support regarding statistics provided by Denis Lacerte, adviser at the Centre de Recherche Universitaire pour les Jeunes et les Familles (CRUJeF).

## Author contributions

**Conceptualization:** Genevieve Piche, Aude Villatte.

**Formal analysis:** Mireille Jasmin, Genevieve Piche.

**Funding acquisition:** Genevieve Piche.

**Investigation:** Genevieve Piche.

**Methodology:** Mireille Jasmin, Genevieve Piche, Aude Villatte, Marianne Fournier-Marceau, Darryl Maybery.

**Project administration:** Genevieve Piche, Marianne Fournier-Marceau.

**Resources:** Genevieve Piche.

**Software:** Genevieve Piche.

**Supervision:** Genevieve Piche, Marie-Ève Clément, Marianne Fournier-Marceau.

**Validation:** Mireille Jasmin, Genevieve Piche, Aude Villatte, Andrea Reupert, Marie-Ève Clément, Anne Dorothee Muller, Marianne Fournier-Marceau, Darryl Maybery.

**Visualization:** Mireille Jasmin, Genevieve Piche, Aude Villatte, Andrea Reupert, Marie-Ève Clément, Anne Dorothee Muller, Marianne Fournier-Marceau, Darryl Maybery.

**Writing – original draft:** Mireille Jasmin, Genevieve Piche, Marianne Fournier-Marceau.

**Writing – review & editing:** Mireille Jasmin, Genevieve Piche, Aude Villatte, Andrea Reupert, Marie-Ève Clément, Anne Dorothee Muller, Marianne Fournier-Marceau, Darryl Maybery, Marie-Hélène Morin, Stéphane Richard-Devantoy.

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
