## [Decision Letter · Decision Letter 0]

29 Dec 2025

Dear Dr. Piche,

We look forward to receiving your revised manuscript.

Kind regards,

Gerard Hutchinson, MD

Academic Editor

PLOS One

Journal Requirements:

2. We note you have included a table to which you do not refer in the text of your manuscript. Please ensure that you refer to Table 3 in your text; if accepted, production will need this reference to link the reader to the Table.

3. In this instance it seems there may be acceptable restrictions in place that prevent the public sharing of your minimal data. However, in line with our goal of ensuring long-term data availability to all interested researchers, PLOS’ Data Policy states that authors cannot be the sole named individuals responsible for ensuring data access (http://journals.plos.org/plosone/s/data-availability#loc-acceptable-data-sharing-methods ).

Reviewers' comments:

Reviewer's Responses to Questions

**Comments to the Author**

1. Is the manuscript technically sound, and do the data support the conclusions?

Reviewer #1: Yes

2. Has the statistical analysis been performed appropriately and rigorously?

Reviewer #1: Yes

3. Have the authors made all data underlying the findings in their manuscript fully available?

Reviewer #1: Yes

4. Is the manuscript presented in an intelligible fashion and written in standard English?

Reviewer #1: Yes

Reviewer #1: Thanks for affording me the opportunity to review this important paper, which examines psychiatrists engagement in FFP in context of parental mental illness. While the sample is small, this is noted in limitations and the authors are correct in indicating that there is limited understanding of psychiatrists' perspectives. It would have been good to see more detail about the adaption of the FFMHPQ and its psychometric properties although i also note that the author has published this detail elsewhere.

**Do you want your identity to be public for this peer review?** For information about this choice, including consent withdrawal, please see our Privacy Policy

Reviewer #1: No

---

## [Author Response · Author response to Decision Letter 1]

28 Jan 2026

We are grateful to the Academic Editor for the thoughtful and valuable comments. Please find our detailed response below.

First Editor’s comment:

1. We note that there were multiple versions of Manuscript in your submission's file inventory. Please double check your submission file inventory retaining the most recent version(s) for editorial review and let us know if any files appear to be outdated or absent.

Response to Editor’s comment:

We tried removing other versions of the manuscript on the portal, without success – an error message keeps us from submitting without all versions included.

Second Editor’s comment:

If there are ethical or legal restrictions on sharing a de-identified data set, please explain them in detail (e.g., data contain potentially identifying or sensitive patient information, data are owned by a third-party organization, etc.) and who has imposed them (e.g., a Research Ethics Committee or Institutional Review Board, etc.). Please also provide contact information for a data access committee, ethics committee, or other institutional body to which data requests may be sent.

Please update your Data Availability statement in the submission form accordingly."

Response to Editor’s comment:

Thank you for your email and for your comments regarding the data availability statement for our manuscript. After carefully reviewing our ethical approval documents, we unfortunately do not have the required participant consent to make the full dataset publicly available, even in de-identified form. Public data sharing was not included in our original information and consent form, and the Research Ethics Committees of the Université du Québec en Outaouais and of the Centre intégré universitaire de santé et de services sociaux (CIUSSS) de la Capitale-Nationale have therefore restricted us from making these data publicly accessible.

In line with PLOS’s guidance that data requests should be directed to a non-author institutional point of contact, we propose the university’s Ethics Committee as a neutral, non-author institutional contact for controlled access to the data, even though data management is not part of its formal mandate. Data requests may be addressed to:

Université du Québec en Outaouais Ethics Committee

Email: comite.ethique@uqo.ca

Accordingly, we propose the following Data Availability statement:

“Data cannot be shared publicly because participants did not provide consent for open data sharing, the dataset contains potentially identifying and sensitive information, and the Research Ethics Committee has restricted public access to the dataset. Data are available from the Université du Québec en Outaouais Ethics Committee (contact via comite.ethique@uqo.ca) for qualified researchers who meet the criteria for access to confidential data.”

Please let us know if further clarification or adjustments are needed.

Best regards,

Geneviève Piché

---

## [Editor Report · Decision Letter 1]

29 Jan 2026

Exploring Psychiatrists' Perspectives on Supporting Parents with Mental Health Challenges: A Mixed-Methods Study

PONE-D-25-51780R1

Dear Dr. Piche

We’re pleased to inform you that your manuscript has been judged scientifically suitable for publication and will be formally accepted for publication once it meets all outstanding technical requirements.

Kind regards,

Gerard Hutchinson, MD

Academic Editor

PLOS One
---

## [Editor Report · Acceptance letter]

PONE-D-25-51780R1

PLOS One

Dear Dr. Piche,

I'm pleased to inform you that your manuscript has been deemed suitable for publication in PLOS One. Congratulations! Your manuscript is now being handed over to our production team.

Kind regards,

on behalf of

Dr. Gerard Hutchinson

Academic Editor

PLOS One